# Higher Circulating Levels of Neutrophils and Basophils Are Linked to Myopic Retinopathy

**DOI:** 10.3390/ijms24010080

**Published:** 2022-12-21

**Authors:** Jinyan Qi, Wei Pan, Ting Peng, Ling Zeng, Xiaoning Li, Zhongping Chen, Zhikuan Yang, Heping Xu

**Affiliations:** 1Aier School of Ophthalmology, Central South University, Changsha 410000, China; 2Aier Institute of Optometry and Vision Science, Changsha 410000, China; 3Hunan Province Optometry Engineering and Technology Research Center, Changsha 410000, China; 4Hunan Province International Cooperation Base for Optometry Science and Technology, Changsha 410000, China; 5Health Management Center, The Third Xiangya Hospital, Central South University, Changsha 410000, China; 6Changsha Aier Eye Hospital, Changsha 410000, China; 7Aier School of Optometry and Vision Science, Hubei University of Science and Technology, Xianning 437100, China; 8Wellcome-Wolfson Institute for Experimental Medicine, School of Medicine, Dentistry and Biomedical Sciences, Queen’s University Belfast, Belfast BT9 7BL, UK

**Keywords:** blood test, immune cells, myopia, retinal degeneration, neutrophils, basophils, platelets

## Abstract

This retrospective study investigated circulating immune cell alteration in patients with myopic retinopathy. Blood test results and demographic and ocular information of 392 myopic patients and 129 emmetropia controls who attended Changsha Aier Eye Hospital from May 2017 to April 2022 were used in this study. Compared with emmetropia, the percentages of neutrophils and basophils and neutrophil/lymphocyte ratio were significantly higher in myopic patients, whereas the percentages of monocytes and lymphocytes and the counts of lymphocytes and eosinophils were significantly lower in myopic patients. After adjusting for age and hypertension/diabetes, the difference remained. Interestingly, the platelet counts were significantly lower in myopic patients after the adjustments. Further subgroup analysis using multivariable linear regression showed that higher levels of neutrophils, neutrophil/lymphocyte ratio, and platelet/lymphocyte ratio, lower levels of monocytes, eosinophils, lymphocytes, and platelets, were related to myopic peripheral retinal degeneration (mPRD) and posterior staphyloma (PS). A higher level of basophils was linked to myopic choroidal neovascularization (mCNV). Our results suggest that higher levels of circulating neutrophils and neutrophil/lymphocyte ratio, lower monocytes, eosinophils, lymphocytes, and platelets are related to mild myopic retinopathy. A higher level of circulating basophils is related to the severe form of myopic retinopathy, such as mCNV.

## 1. Introduction

Myopia has become a serious health problem worldwide, particularly in Asia [1,2,3,4,5,6,7,8,9]. In 2000, 22.9% of the world’s population had myopia, and 2.7% were high myopia [10]. The numbers are expected to increase to ~50% and 10%, respectively, by 2050 [10]. With the increasing prevalence of myopia, in particular high myopia, the number of people with myopic retinopathy is expected to rise significantly. High myopia can lead to degenerative changes in the neuronal retina, including peripheral retinal degeneration (mPRD), posterior staphyloma (PS), myopic rhegmatogenous retinal detachments (mRRD), myopic maculopathy (MM) and myopic choroidal neovascularization (mCNV), collectively known as “myopic retinopathy”. It has been reported that the incidence of myopic retinopathy in high-myopic populations ranges from 4.27% to 61.7% in different studies [8,11,12,13]. Age is a risk factor for myopic retinopathy [14]. The majority of myopic retinopathy is sight-threatening, yet the underlying mechanisms remain largely unknown. Inflammation is known to play a role in the development of myopia. Patients with autoimmune and inflammatory diseases, such as juvenile chronic arthritis, uveitis, type 1 diabetes mellitus or systemic lupus erythematosus have a higher incidence of myopia [15,16,17]. Higher intraocular levels of inflammatory mediators such as IL-6, VEGF, ICEM-1 [18], and complement proteins [19] have been detected in eyes with myopic retinopathy, suggesting that inflammation may also be involved in myopia-mediated retinal degeneration. Indeed, inflammation critically contributes to retinal degeneration caused by different reasons, such as age (i.e., age-related macular degeneration), diabetes (diabetic retinopathy), glaucoma, and genetic mutation (e.g., retinitis pigmentosa) [20]. 

Myopic retinopathy can be caused by progressive axial length (AL) elongation and/or the degeneration or functional alteration of retinal neurons resulting from abnormal light processing [21,22]. Axial length, myopic diopter, duration of myopia, and age of myopic onset are established risk factors and the strongest predictors for myopic retinopathy [23,24]. The altered circulating immune cell activation may participate in choroidal/scleral remodeling, thereby contributing to the progression of myopia and the development of myopic retinopathy. Inside the eye, the degenerative/dead neurons are handled by retinal microglia and the intraocular complement system when the blood-retinal barrier is intact [20]. As the disease progresses, circulating immune cells may be recruited to assist microglia to remove debris [20]. Therefore, the activation level of systemic immune cells may influence myopic retinopathy. 

The aim of this retrospective case-control study was to investigate the link between systemic immune cell alterations and myopic retinopathy. We also evaluated the relationship between AL, myopic diopter, myopic duration, age of myopic onset, and circulating immune cells.

## 2. Results

### 2.1. Clinical Characteristics

There was no significant difference in gender distribution, history of allergy disease, history of drug allergy, surgical history, or BMI between myopic patients and controls. However, the median age of myopic patients was significantly younger than controls (31 years vs. 43 years, *p* = 0.000, Table 1). A significantly smaller number of participants in the myopia group had a history of hypertension or diabetes compared to the control group (*p* = 0.000, Table 1). 

Among the 392 myopic patients, 96 (24.49%) were simple high myopia (SHM), 60 (15.31%) had mPRD, 58 (14.80%) had PS, 86 (21.94%) had mRRD, 37 (9.44%) had MM and 55 (14.03%) had mCNV (Table 1). The age, gender distribution, history of hypertension or diabetes, surgical history, history of drug allergy, and BMI in subgroups of the myopic patients were significantly different from the controls. Patients with SHM, mPRD, and PS were younger, whereas patients with MM and mCNV were older than controls. There was a significantly higher number of females in the PS group but fewer in the mRRD group. The number of people with a surgical history in the SHM group was less than the controls. More people in the MM and mCNV groups had a history of drug allergies. Regarding the history of systemic disease, significantly fewer people were reported in SHM, mPRD, and PS groups. BMI was lower in mPRD and PS. No significant differences were observed in the history of allergic disease (Table 1).

### 2.2. Circulating Immune Cells in Myopic Patients and Controls

Compared with emmetropia control, the percentages of neutrophils and basophils were significantly higher in myopic patients, whereas the percentage of monocytes and lymphocytes, lymphocyte, and eosinophil counts were significantly lower in myopic patients. The neutrophil/lymphocyte ratio (NLR) in myopia was significantly higher than in controls (all *p* < 0.05). Other parameters, including WBC, neutrophil, monocyte, basophil and platelet counts, PLR, LMR, and the percentage of eosinophils, did not differ significantly between the two groups. After adjusting for age and history of systemic disease, the differences remained. Interestingly, platelet counts were significantly lower in myopic patients after the adjustments (Table 2). 

### 2.3. Correlation between Immune Cell Parameters and Axial Length (AL), Myopic Diopter, Myopic Duration, Age of Myopic Onset

Of the 521 participants, information on AL and myopic diopter was available in 415 and 482, respectively. We then investigated their correlations with circulating immune cells. A weak positive correlation was observed between AL and NLR (r = 0.104, *p* = 0.035). Myopic diopter was positively correlated with lymphocyte and platelet counts but negatively correlated with the percentage of basophils and NLR (Table 3). The myopic duration was positively correlated with eosinophils and basophils but negatively correlated with platelets (Table 3). We did not observe any correlations between the age of myopic onset and circulating immune cells (Table 3). 

Taken together, our results suggest that the longer myopic duration, the higher circulating basophils and eosinophils, and the higher myopia, the lower platelet counts. However, these were weak correlations, and their clinical significance may be limited. Further studies on the role of eosinophils/basophils and platelets in the development and progression of myopia will be needed.

### 2.4. Circulating Immune Cells in Subgroups of Myopic Patients and Controls

Further analysis of blood parameters between emmetropia control and subgroups of myopic patients showed that WBC counts were significantly lower in MM. Patients with SHM, mPRD, and PS had significantly higher levels of neutrophils but lower levels of lymphocytes and eosinophils. NLR was also higher in these patients (Table 4). The platelet count was significantly lower in mRRD. Patients with mCNV had significantly higher levels of basophils (Table 4). 

After adjusting for age and history of systemic disease, the differences between control and MM disappeared (Table 4). The differences between controls and other subgroups of myopia (SHM, mPRD, PS, mRRD, and mCNV) remained (Table 4). Platelet counts were significantly lower in mPRD and PS (in addition to mRRD) after the adjustment (Table 4). The difference in NLR between mRRD and controls became statistically significant after the adjustment (Table 4).

Taken together, our results suggest that high myopic patients without retinal degeneration (i.e., SHM) or mild retinal degenerations (e.g., mPRD, PS) appear to have higher levels of neutrophils but lower levels of monocytes, eosinophils, and lymphocytes. Whereas patients with severe myopic retinopathy (e.g., mCNV) appear to have a higher level of basophils.

## 3. Discussion

In this study, we show that myopic patients had higher levels of circulating neutrophils but lower levels of monocytes, eosinophils, lymphocytes, and platelets compared to emmetropia. The NLR in myopia was significantly higher than that in controls. These differences are mainly in myopic patients without retinopathy (i.e., SHM) or mild myopic retinopathy (e.g., mPRD and PS). In severe myopic retinopathy groups, patients with mCNV had a significantly higher level of basophils compared to emmetropic controls. Our results affirmed the involvement of the systemic immune system in myopia. More importantly, we uncovered the link between myopic retinopathy and altered circulating immune cells.

Previous studies reported a higher incidence of myopia in people with inflammatory or autoimmune diseases, suggesting a role of systemic immune system alteration in the development of myopia. In this study, we excluded the participants with systemic inflammatory or autoimmune diseases. This study design allowed us to investigate the link between basal-level circulating immune cell profiles and myopia. Old age and chronic diseases such as diabetes can lead to low levels of inflammation known as “inflammaging” [25] and “para-inflammation” [26]. We found that myopic patients had higher levels of neutrophils and basophils but lower levels of monocytes, eosinophils, lymphocytes, and platelets and higher levels of NLR after adjusting for age and history of systemic chronic diseases. Further study on the functional alteration of these cells will help to understand the causal role of inflammation in the development of myopia. 

The SHM patients had spherical equivalent refraction ≤ −6.0 D but no clinically detectable retinal/choroidal pathologies. Previous studies have shown that the thickness of the choroid and retina in SHM was reduced [27]. In mPRD, neurons in the peripheral retina degenerate, leading to “lattice” structure formation (Lattice degeneration) or retinal hole [28]. Posterior staphyloma is due to the weakening of the sclera and subsequent degeneration of the choroid and retina in myopic eyes [29]. The cellular and molecular mechanisms that underly retinal, choroidal, and scleral degeneration in these conditions remain unknown. Abnormal light processing can affect choroidal/scleral remodeling, known as the retina-choroid-sclera pathway [30]. The progressive thinning of the choroid and sclera in pathological myopia may lead to the development of posterior staphyloma [27], lacquer cracks and chorioretinal atrophy [25]. Whereas, decreased choroidal perfusion may lead to outer retinal ischemia and the development of MM and mCNV [26]. 

The peripheral retina and juxtapupillary area are the weak points of retinal protection due to their close connections with the pars plana and choroid [31], which allow easy access of circulating immune cells and blood-borne pathogens to the retina. The circulating immune cells may migrate to the choroidal layer, where they can modulate the immune response in the peripheral retina and juxtapupillary area leading to regional retinal degeneration and myopic retinopathy. 

We found higher circulating levels of neutrophils in patients with no or mild myopic retinopathy. Higher neutrophils have been reported in other retinal degenerative conditions such as primary glaucoma [32], AMD [33], and diabetic retinopathy [34]. Neutrophil-derived elastase and proteinase are known to play a role in different degenerative and inflammatory diseases by their proteolysis of collagen-IV and elastin of the extracellular matrix metalloproteinase [35]. In addition, neutrophil extracellular traps (NETs) can target senescent vasculature for tissue remodeling in retinopathy [36]. Neutrophils may participate in the remodeling of scleral, choroidal, and retinal degeneration in myopic conditions (e.g., SHM, mPRD, and PS) by releasing abnormal levels of elastase/proteinase or the formation of NETs. Further functional investigation will be needed to elucidate the underlying mechanism.

Interestingly, the platelet counts were significantly lower in patients with mild (mPRD, PS, and mRRD) but not severe (MM and mCNV) myopic retinopathy. Platelets are known to have a central role in the pathogenesis of a diverse array of inflammatory diseases, such as atherosclerosis, sepsis-induced organ damage, and rheumatoid arthritis [37]. Platelets can store various immunomodulatory molecules, including TGFβ, interleukin-1, platelet-derived growth factor, and CC-chemokine ligand five, that can markedly affect immune responses. Platelets express and secrete CD40 and CD154, which can affect both dendritic cell maturation and T cell activation [37]. The lower platelet counts may lead to higher circulating levels of chemokines, cytokines, and growth factors, which may affect immune response in the sclera, choroid, and retina in myopic retinopathy. This will be an important question for future investigation.

Myopic CNV is the growth of abnormal blood vessels from the choroid into the retina, including the macula [38,39]. The disease can be treated with intravitreal injection of VEGF inhibitors, although the pathogenesis of the condition remains poorly defined [39]. We found that mCNV patients had a higher circulating level of basophils compared to emmetropia controls. Basophils are the least common granulocytes, accounting for ~0.5% of circulating leukocytes in normal conditions. Their roles in health and disease have been overlooked in the past, but recent studies have shown that they are involved in a broad spectrum of human diseases [40]. In addition to allergic responses, basophils are critically involved in autoimmune diseases, cancer, tissue repair, and fibrosis by releasing cytokines such as IL-4 and IL-13 that can modulate Th2 and M2-type immune responses [40,41]. Basophils-mediated Th2/M2-type immune response may promote choroidal neovascularization and the development of mCNV, although experimental studies will be needed to confirm this. 

We did not observe any significant immune cell alterations in MM. However, this does not mean that circulating immune cells are not involved in this condition. Further functional analysis of different immune cells will be needed to fully understand their involvement in the development and progression of the spectrum of myopic retinopathy. 

Major strengths of this study include the matched emmetropia controls, different subgroups of myopic retinopathy, comprehensive demographic information and blood test results, and the exclusion of participants with inflammatory diseases. There are several limitations in the study. First, the sample size in some myopic retinopathy groups is relatively small (e.g., *n* = 37 in MM). Second, there was a significant difference in age between myopic patients and controls. Third, medical examinations were conducted at a single time point. Consequently, some patients diagnosed with simple myopia might develop myopic retinopathy later. A follow-up study on these participants will add valuable information on the role of altered immune cells in myopic retinopathy. Furthermore, the study was conducted in a single center; thus, the results only reflect the local population. Further multicenter studies will be needed to verify the results in other populations. It is worth noting that the single-center study is also a strength of the study as test procedures were consistent between participants, and this increases the robustness of the results despite the small sample size. Finally, the study only examined the number but not the function of circulating immune cells. Therefore, we do not know immune cell alteration may affect myopia and myopic retinopathy.

## 4. Materials and Methods

### 4.1. Clinical Data Collection

This is a retrospective case-control study. We analyzed the clinical records of 392 myopic patients who underwent lens replacement surgery or sought medical assistance due to myopic retinopathy and 129 healthy emmetropia (controls) who had routine health checkups at Changsha Aier Eye hospital from May 2017 to April 2022. This study was performed in accordance with the Declaration of Helsinki and was approved by the Institutional Review Board (IRB) of Changsha Aier Eye Hospital (Ref: 2021KYPJ008). 

Myopic patients were confirmed by myopic diopter (<−1.0D) and axial length (AL ≥ 24 mm). Myopic eyes were further divided into below six subgroups: 

Simple high myopia (SHM, *n* = 96), spherical equivalent refraction ≤ −6.0 D or AL ≥ 26.5 mm, and without any myopic retinopathy (Appendix A Figure A1A).

mPRD (*n* = 60), myopia with lattice degeneration, white appearance without pressure, paving stone, pigmentary degenerations, or retinal holes in fundus examination (Appendix A Figure A1B).

PS (*n* = 58), B-scan ocular ultrasound examination showing an outpouching of the wall of the eye that has a radius of curvature that is less than the surrounding curvature of the wall of the eye in myopic patients (Appendix A Figure A1C).

mRRD (*n* = 86), rhegmatogenous retinal detachment in myopic patients confirmed by fundus examination (Appendix A Figure A1D).

Myopic maculopathy (MM, *n* = 37), macular retinal detachment, or macular retinoschisis in myopic patients confirmed by fundus examination and/or optical coherence tomography (OCT) (Appendix A Figure A1E).

mCNV (*n* = 55), choroidal neovascularization confirmed by fundus fluorescein angiography (FFA) and/or OCT and optical coherence tomography angiography (OCTA) in myopic patients (Appendix A Figure A1F).

For healthy controls, emmetropia was confirmed by uncorrected visual acuity (UCVA = 1.0) and/or myopic diopter (±0.5D). No fundus abnormalities (Appendix A Figure A1G).

Patients with a history of retinal surgery or laser treatment, pregnancy, active inflammation or autoimmune diseases (e.g., acute/chronic infection, active rheumatoid arthritis, multiple sclerosis, hyperthyroidism, undergoing immunosuppressive therapy, chemotherapy, etc.), or other retinal diseases (e.g., glaucoma, diabetic retinopathy) were excluded from the study. The below information was extracted from the medical record of each participant: age, gender, body mass index (BMI), family history of myopia, blood test results, ocular examination results, demographic information, etc.

### 4.2. Blood Test

The following parameters were collected from the blood test results: white blood cells count (WBC), the counts of neutrophils, lymphocytes, monocytes, eosinophils, basophils, platelets, platelet-to-lymphocyte ratio (PLR), lymphocyte-to-monocyte ratio (LMR) and neutrophil-to-lymphocyte ratio (NLR), the percentages of neutrophils, lymphocytes, monocytes, eosinophils, and basophils.

### 4.3. Ocular Information

Each myopic patient received a complete ophthalmic examination at the hospital visit. These include AL, myopic diopter, best-corrected visual acuity (BCVA), UCVA, slit-lamp examination, intraocular pressure measurement, fundus stereoscopic biomicroscopy, FFA, OCT, and OCTA, and B-scan ocular ultrasound. The healthy emmetropia had gone BCVA, UCVA, AL, myopic diopter measurement, slit-lamp examination, and fundus stereoscopic biomicroscopy. 

### 4.4. Demographic Information

The following information was extracted from the medical records of each participant: age, gender, myopic duration, age of myopic onset, BMI, the use of other medications (e.g., to control hypertension, diabetes, aspirin, hormone supplements, etc.), family history of myopia, contact lens usage, smoking history, other eye diseases, history of systemic disease (e.g., hypertension, diabetes), history of allergy disease, autoimmune diseases, surgical history, history of drug allergy, history of vaccinations.

### 4.5. Statistical Analysis

Data were analyzed using the SPSS 20.0 software. The normal distribution of continuous variables was evaluated by the Kolmogorov-Smirnov test. The continuous variables were compared by the Student’s *t*-test or the Mann–Whitney *U* test. The chi-square test was used to compare categorical variables. We used the least squares (LS) criterion to determine the regression line. The least squares mean (LSM) ± standard error in each group and the LSM difference (95% CI) between myopia and control group were calculated using multivariable linear regression adjusted by age and history of systemic disease. The correlation between blood immune cells and AL, myopic diopter, the myopia of duration, and the age of myopic onset were assessed using the Pearson or Spearman correlation test based on normality. *p* < 0.05 was considered statistically significant.

## 5. Conclusions

In this study, we found that higher levels of neutrophils and NLR, and lower levels of monocytes, eosinophils, lymphocytes, and platelets are related to the development of myopia and the progression of the disease to mild myopic retinopathy. Whereas a higher level of basophils is related to the severe form of myopic retinopathy such as mCNV. Our results suggest a link between altered innate and adaptive immunity and the development and progression of myopia and myopic retinopathy.

## Figures and Tables

**Table 1 ijms-24-00080-t001:** Demographic and clinical characteristics of study participants.

Participant Characteristics	Control *n* = 129	Myopia *n* = 393	Subgroups of Myopia					
			SHM *n* = 96	mPRD *n* = 60	PS *n* = 58	mRRD *n* = 86	MM *n* = 37	mCNV *n* = 55
Age, Median (IQR) ^a^	43 (33–54)	**31 (23–46)** ******	**24 (21–30) ****	**24 (20–30) ****	**27 (24–34) ****	40 (31–51)	56 (51–68) **	50 (36–59) *
Female (%) ^b^	58.91	61.73	69.79	63.33	79.31 **	36.05 **	70.27	61.82
History of allergic disease (%) ^b^	17.05	15.24	12.00	13.95	25.00	17.54	15.79	8.57
History of drug allergy (%) ^b^	1.55	4.42	1.04	6.66	0.00	4.69	10.71 *	9.09 *
History of hypertension/diabetes (%) ^b^	20.90	**8.90** ******	**0.00** ******	**0.00** ******	**0.00** ******	11.60	32.40	23.60
Surgical history (%) ^b^	27.13	25.20	**15.63** *****	15.00	22.41	36.20	43.33	32.73
BMI (Mean ± SD) ^c^	23.72 ± 3.63	22.94 ± 3.60	22.64 ± 5.27	**21.27 ± 3.86** ******	**20.40 ± 2.78** ******	23.42 ± 3.26	23.45 ± 3.01	23.77 ± 3.29

^a^ Mann-Whitney U test; ^b^ Chi-square test; ^c^ Independent sample *t*-test. * *p* < 0.05; ** *p* < 0.01. **Bold** indicating *p*-value is statistically significant compared to controls. SHM: simple high myopia; mPRD: myopia with peripheral retinal degeneration; PS: posterior staphyloma; mRRD: myopic rhegmatogenous retinal detachment; MM: myopic maculopathy; mCNV: myopic choroidal neovascularization; BMI: body mass index.

**Table 2 ijms-24-00080-t002:** Blood immune cells in myopic patients and controls.

Blood Immune Cells LSM ± SE	Control *n* = 129	Myopia *n* = 392	*p* Value
WBC 10^9^/L	6.21 ± 0.16	6.07 ± 0.12	0.41
Neutrophils (%)	56.24 ± 0.88	58.76 ± 0.66	**0.005**
Lymphocytes (%)	34.85 ±0.81	32.91 ± 0.61	**0.02**
Monocytes (%)	5.88 ±0.15	5.48 ± 0.11	**0.008**
Eosinophils (%)	2.45 ±0.18	2.20 ±0.13	0.16
Basophils (%)	0.57 ±0.03	0.67 ± 0.03	**0.007**
Neutrophils (10^9^/L)	3.54 ±0.13	3.63 ±0.10	0.53
Lymphocytes (10^9^/L)	2.11 ± 0.06	1.94 ± 0.04	**0.004**
Monocytes (10^9^/L)	0.37 ± 0.04	0.35 ±0.03	0.65
Eosinophils (10^9^/L)	0.19 ± 0.01	0.15 ± 0.01	**0.002**
Basophils (10^9^/L)	0.05 ±0.00	0.04 ± 0.00	0.28
Platelets (10^9^/L)	225.70 ±5.35	213.50 ±4.03	**0.02**
PLR	111.29 ± 4.28	117.84 ±3.23	0.13
LMR	6.28 ±0.20	6.27 ± 0.15	0.98
NLR	1.74 ±0.11	2.02 ±0.08	**0.01**

*p*-value in **bold** indicating statistically significant by multivariable linear regression analysis adjusted for age and history of hypertension and/or diabetes. LSM: Least-Squares Means; SE: Standard Error; WBC: white blood cells count; PLR: platelet-to-lymphocyte ratio; LMR: lymphocyte-to-monocyte ratio; NLR: neutrophil-to-lymphocyte ratio.

**Table 3 ijms-24-00080-t003:** Correlations between blood immune cells and myopic parameters.

Blood Immune Cells	AL *n* = 415		Myopic Diopter *n* = 482		Myopia Duration *n* = 311		Age of Myopic Onset *n* = 311	
	r	*p* ^a^	r	*p* ^a^	r	*p* ^b^	r	*p* ^b^
WBC 10^9^/L	0.018	0.716	0.028	0.525	−0.026	0.648	−0.006	0.912
Neutrophils (%)	0.063	0.198	−0.062	0.160	−0.045	0.424	0.055	0.335
Lymphocytes (%)	−0.065	0.187	0.057	0.197	0.027	0.631	−0.015	0.793
Monocytes (%)	−0.052	0.289	0.056	0.202	−0.048	0.402	−0.106	0.061
Eosinophils (%)	0.013	0.796	0.021	0.636	0.126	**0.027**	−0.043	0.445
Basophils (%)	0.019	0.695	−0.09	**0.039**	0.134	**0.018**	−0.099	0.083
Neutrophils (10^9^/L)	0.051	0.304	−0.012	0.779	−0.031	0.585	0.019	0.735
Lymphocytes (10^9^/L)	−0.062	0.208	0.086	**0.049**	−0.029	0.606	−0.047	0.408
Monocytes (10^9^/L)	−0.044	0.372	0.028	0.521	−0.043	0.445	−0.09	0.111
Eosinophils (10^9^/L)	0.005	0.917	0.081	0.066	0.112	**0.049**	−0.067	0.236
Basophils (10^9^/L)	−0.046	0.345	0.041	0.355	0.072	0.203	−0.109	0.054
Platelet (10^9^/L)	−0.027	0.575	0.093	**0.034**	−0.139	**0.014**	−0.096	0.090
PLR	0.044	0.373	−0.044	0.313	−0.048	0.398	−0.052	0.363
LMR	−0.009	0.856	0.006	0.885	0.053	0.354	0.053	0.351
NLR	0.104	**0.035**	−0.089	**0.043**	−0.031	0.588	0.024	0.672

^a^ Pearson correlation test; ^b^ Spearman correlation test. **Bold** indicating *p*-value is statistically significant compared to the control. WBC: white blood cells count; PLR: platelet-to-lymphocyte ratio; LMR: lymphocyte-to-monocyte ratio; NLR: neutrophil-to-lymphocyte ratio; AL: axial length.

**Table 4 ijms-24-00080-t004:** Multivariable linear regression analysis of blood immune cells between emmetropic controls and the subgroups of myopic patients.

Blood Immune Cells (LSM Difference (95% CI))	SHM *n* = 96	mPRD *n* = 60	PS *n* = 58	mRRD *n* = 86	MM *n* = 37	mCNV *n* = 55
WBC 10^9^/L	−0.44 (−0.91, 0.03)	−0.35 (−0.88, 0.17)	−0.46 (−0.98, 0.05)	0.10 (−0.33, 0.53)	−0.30 (−0.90, 0.29)	0.31 (−0.19, 0.81)
Neutrophils (%)	**4.58 (2.08, 7.08) ****	**5.38 (2.58, 8.18) ****	**5.80 (3.05, 8.55) ****	1.19 (−1.10, 3.49)	−0.95 (−4.14, 2.25)	1.15 (−1.52, 3.82)
Lymphocytes (%)	**−3.26 (−5.59, −0.92) ****	**−4.17 (−6.78, −1.56) ****	**−4.40 (−6.97, −1.83) ****	−0.98 (−3.13, 1.16)	0.57 (−2.42, 3.55)	−0.96 (−3.46, 1.53)
Monocytes (%)	**−0.51 (−0.94, −0.09) ***	**−0.69 (−1.17, −0.21) ****	**−0.61 (−1.08, −0.13) ***	−0.35 (−0.74, 0.05)	0.05 (−0.50, 0.59)	−0.38 (−0.84, 0.07)
Eosinophils (%)	**−0.76 (−1.26, −0.25) ****	**−0.67 (−1.24, −0.11) ***	**−0.88 (−1.43, −0.32) ****	0.09 (−0.37, 0.55)	0.21 (−0.43, 0.86)	0.08 (−0.46, 0.62)
Basophils (%)	0.08 (−0.02, 0.18)	0.10 (−0.01, 0.21)	0.09 (−0.01, 0.20)	0.04 (−0.05, 0.13)	0.08 (−0.04, 0.21)	**0.18 (0.07, 0.28) ****
Neutrophils (10^9^/L)	0.02 (−0.36, 0.41)	0.13 (−0.30, 0.55)	0.11 (−0.31, 0.53)	0.14 (−0.21, 0.49)	−0.23 (−0.72, 0.26)	0.27 (−0.14, 0.68)
Lymphocytes (10^9^/L)	**−0.35 (−0.52, −0.19) ****	**−0.36 (−0.55, −0.18) ****	**−0.44 (−0.62, −0.25) ****	−0.03 (−0.18, 0.12)	−0.07 (−0.28, 0.14)	0.03 (−0.15, 0.20)
Monocytes (10^9^/L)	0.02 (−0.11, 0.15)	−0.08 (−0.22, 0.07)	−0.08 (−0.22, 0.06)	−0.02 (−0.14, 0.10)	−0.00 (−0.17, 0.16)	−0.01 (−0.14, 0.13)
Eosinophils (10^9^/L)	**−0.08 (−0.12, −0.04) ****	**−0.07 (−0.12, −0.03) ****	**−0.08 (−0.13, −0.04) ****	−0.02 (−0.06, 0.02)	−0.03 (−0.08, 0.02)	−0.01 (−0.06, 0.03)
Basophils (10^9^/L)	−0.00 (−0.02, 0.01)	−0.00 (−0.02, 0.01)	−0.00 (−0.02, 0.01)	−0.01 (−0.02, 0.00)	−0.01 (−0.02, 0.01)	0.00 (−0.01, 0.01)
Platelets (10^9^/L)	−5.62 (−21.03, 9.79)	**−18.58 (−35.84, −1.32) ***	**−18.46 (−35.43, −1.49) ***	**−17.27 (−31.44, −3.09) ***	−5.24 (−24.96, 14.48)	−9.59 (−26.07, 6.89)
PLR	**16.39 (4.10, 28.68) ****	10.14 (−3.62, 23.91)	15.54 (2.01, 29.07) *	−0.51 (−11.81, 10.79)	1.97 (−13.76, 17.69)	2.04 (−11.10, 15.19)
LMR	−0.21 (−0.79, 0.38)	−0.26 (−0.91, 0.40)	−0.21 (−0.85, 0.43)	0.22 (−0.32, 0.75)	−0.04 (−0.79, 0.71)	0.22 (−0.41, 0.84)
NLR	**0.34 (0.03, 0.65) ***	**0.44 (0.10, 0.79) ***	**0.48 (0.14, 0.82) ****	**0.29 (0.01, 0.58) ***	−0.06 (−0.46, 0.33)	0.22 (−0.11, 0.55)

* *p* < 0.05; ** *p* < 0.01 compared to emmetropic controls after adjusting for age, gender, history of hypertension/diabetes, surgical history, and BMI with multivariable linear regression analysis. LSM: lease squares mean; SHM: simple high myopia; mPRD: myopia with peripheral retinal degeneration; PS: posterior staphyloma; mRRD: myopic rhegmatogenous retinal detachment; MM: myopic maculopathy; mCNV: myopic choroidal neovascularization.

## Data Availability

The authors declare that all data supporting the findings are available within the paper.

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
