# Peer review of "Higher Circulating Levels of Neutrophils and Basophils Are Linked to Myopic Retinopathy"

_ijms, 2022, doi:10.3390/ijms24010080_

Round 1

Reviewer 1 Report

The paper reports the findings of a retrospective study that aims to investigate the link between systemic immune cell alterations and myopic retinopathy. The premise for the work is that differences in inflammatory status may be altering the pathology of the structures of the eye associated with myopia and in particular the development of high myopia.

The paper is written clearly and identifies possible associations between circulating inflammatory cells and the classification/sub-type of eye pathology experienced. The authors list several limitations of the work – some of which could be a strengths e.g. the single centre study could be seen as a strength as it is likely that test procedures and their interpretation is consistent between patients, increasing the robustness despite the small sample size. 

A bigger limitation of the study and which is not articulated is that this is a single point or cross-sectional analysis with a relatively small sample size, i.e. only a single time-point blood measure is described and there is no link made in the article between what might constitute inflammation and how this might vary within an individual over time, and similarly single timepoint eye assessment. Addressing these aspects – e.g. by recalling and following-up the cases would strengthen the study as otherwise it is interesting but largely descriptive.

Line 62 – reference or explain the link between abnormal light processing and the death of neurons 

Line 197 – explain, evidence, the link made between the ‘immunologically weak points’ due to the connection with the pars plana and choroid

Line 181 - There is a lack of clarity re the inclusion/exclusion criteria re systemic inflammatory diseases.

Author Response

Reviewer 1: The paper reports the findings of a retrospective study that aims to investigate the link between systemic immune cell alterations and myopic retinopathy. The premise for the work is that differences in inflammatory status may be altering the pathology of the structures of the eye associated with myopia and in particular the development of high myopia.

Comment 1: The paper is written clearly and identifies possible associations between circulating inflammatory cells and the classification/sub-type of eye pathology experienced. The authors list several limitations of the work – some of which could be a strengths e.g. the single centre study could be seen as a strength as it is likely that test procedures and their interpretation is consistent between patients, increasing the robustness despite the small sample size. 

Response: We thank the reviewer for this comment and agree that single centre study also offers a strength in terms of inter-centre measurement variation. We have revised the manuscript accordingly (last paragraph of Discussion, lines 262-264). 

Comment: A bigger limitation of the study and which is not articulated is that this is a single point or cross-sectional analysis with a relatively small sample size, i.e. only a single time-point blood measure is described and there is no link made in the article between what might constitute inflammation and how this might vary within an individual over time, and similarly single timepoint eye assessment. Addressing these aspects – e.g. by recalling and following-up the cases would strengthen the study as otherwise it is interesting but largely descriptive.

Response: We agree with the reviewer that a single point of investigation in a relatively small number of participants is a limitation of this study. We have discussed the limitation in the manuscript (number three of the five limitations, lines 258-259). Our future research plan would be to recall the participants and follow up their myopic progression.  

Comment: Line 62 – reference or explain the link between abnormal light processing and the death of neurons 

Response: We previously observed reduced number of retinal neurons in an animal model of myopic retinopathy (PMID: 34830490). Myopic risk genes are expressed in various retinal neurons (PMID: 29808027).  The results suggest that abnormal light processing may damage retinal neurons and induce a retina-to-sclera signalling pathway of myopic progression. We have revised the sentence and it is now read as “Myopic retinopathy can be caused by progressive axial length (AL) elongation and/or the degeneration or functional alteration of retinal neurons resulting from abnormal light processing”. Relevant references have been cited.

Comment: Line 197 – explain, evidence, the link made between the ‘immunologically weak points’ due to the connection with the pars plana and choroid

Response: We thank the reviewer for this comment. Our original description of these two areas as “immunologically weak points” was inappropriate. They are weak points in terms of retinal defence due to the close connection with the pars plan and choroid, which allows the easy access of circulating immune cells and blood-borne pathogens to the retina. We have revised the manuscript accordingly (lines 202-205). The sentence is now read as “The peripheral retina and juxtapupillary area are the weak points of retinal protection due to their close connections with the pars plana and choroid [27], which allow easy access of circulating immune cells and blood-borne patho-gens to the retina”.    

Comment: Line 181 - There is a lack of clarity re the inclusion/exclusion criteria re systemic inflammatory diseases.

Response: We have now clarified the exclusion criteria for systemic inflammatory diseases and included relevant examples. Below sentence was added to the revised manuscript.

“Patients with a history of retinal surgery or laser treatment, pregnancy, active inflammation or autoimmune diseases (e.g., acute/chronic infection, active rheumatoid arthritis, multiple sclerosis, hyperthyroidism, undergoing immunosuppressive therapy, chemotherapy, etc.)”

Reviewer 2 Report

Authors made a good attempt to explain the importance of neutrophils and basophils in the manuscript entitled "Higher circulating levels of neutrophils and basophils are linked to myopic retinopathy" although the study is a retrospective study, authors should have included data related to ocular imaging modalities to make the manuscript more interest and adds merit. 

Author Response

Comment: Authors made a good attempt to explain the importance of neutrophils and basophils in the manuscript entitled "Higher circulating levels of neutrophils and basophils are linked to myopic retinopathy" although the study is a retrospective study, authors should have included data related to ocular imaging modalities to make the manuscript more interest and adds merit

Response: We thank the reviewer for this constructive comment. We have now included representative fundus images from each subgroup of myopic patients (Appendix figure A1).

Reviewer 3 Report

This study has investigated the changes in the levels of immune cell populations in myopic retinopathy vs. emmetropia control subjects. The manuscript provides adequate details about the study subjects, the blood test results demonstrating immune cell levels are convincing, and the data have been logically analyzed by appropriate statistical methods. However, the authors are requested to address the following comments which could eventually strengthen the rationale of this manuscript.

 1.     Provide information on the worldwide prevalence of myopic retinopathy in the introduction.

2.     Please explain how does BMI relate to the pathology of mPRD? Do authors think BMI plays a crucial role in any of the pathologies associated with mPRD?

3.     Although the authors have convincingly acknowledged the limitations of the study, it is important to provide supporting evidence of the expression of upregulating immune cell markers by ELISA.

4.     Discuss the association of scleral, choroidal, and retinal degeneration in relation to myopic retinopathy.

5.     Despite the roles of immune cells have been correlated with autoimmune diseases, please discuss the putative roles of each immune cell with the subgroups of myopia.

6.     Provide the representative images/data of ocular information for the subgroups of myopia patients and emmetropia controls in the supplementary material.  

Author Response

Comment 1: Provide information on the worldwide prevalence of myopic retinopathy in the introduction.

Response: The prevalence of myopic retinopathy varies in different countries, We have included the information in the introduction (lines 48-50)

Comment 2: Please explain how does BMI relate to the pathology of mPRD? Do authors think BMI plays a crucial role in any of the pathologies associated with mPRD?

Response: In our study, patients with mPRD had significantly smaller BMIs compared to controls (Table 1). Since our aim was to understand the link between circulating immune cell alteration and myopic retinopathy and immune cell parameters can be affected by BMI, in our multivariable linear regression analysis, we adjusted for BMI and other potential confounders. Although BMI reflects the overall metabolic states of a person and dysregulated lipid metabolism is known to play a role in various diseases, the link between BMI and myopic retinopathy remains elusive. However, the question is out of the scope of this study.

Comment 3. Although the authors have convincingly acknowledged the limitations of the study, it is important to provide supporting evidence of the expression of upregulating immune cell markers by ELISA.

Response: We agree that additional information on cytokines and chemokines produced by circulating immune cells will provide important information of the functional alteration of relevant immune cells. Due to the nature of the study (i.e., retrospective), we do not have samples for ELISA investigation, unfortunately. However, we have planned future studies to investigate the functional of circulating immune cells in patients with myopic retinopathy

Comment 4. Discuss the association of scleral, choroidal, and retinal degeneration in relation to myopic retinopathy.

Response: We have now discussed the association of scleral, choroidal and retinal degeneration in the context of myopic retinopathy (lines 195-200).

Comment 5. Despite the roles of immune cells have been correlated with autoimmune diseases, please discuss the putative roles of each immune cell with the subgroups of myopia.

Response: We have now discussed the putative roles of immune cells including neutrophils, platelets and basophils in subgroups of myopic retinopathy (lines 214-218; 227-230; 242-244).

Comment 6. Provide the representative images/data of ocular information for the subgroups of myopia patients and emmetropia controls in the supplementary material.  

Response: We thank the reviewer for this constructive comment. We have now included representative fundus images from each subgroup of myopic patients (Appendix figure A1).